# Passive Immunity and Antibody Response Induced by *Toxoplasma gondii* VLP Immunization

**DOI:** 10.3390/vaccines9050425

**Published:** 2021-04-23

**Authors:** Hae-Ji Kang, Min-Ju Kim, Ki-Back Chu, Su-Hwa Lee, Eun-Kyung Moon, Fu-Shi Quan

**Affiliations:** 1Department of Biomedical Science, Graduate School, Kyung Hee University, Seoul 02447, Korea; haedi1202@naver.com (H.-J.K.); mj16441@naver.com (M.-J.K.); ckb421@gmail.com (K.-B.C.); 2Department of Medical Zoology, Kyung Hee University School of Medicine, Seoul 02447, Korea; dltnghk228@nate.com (S.-H.L.); ekmoon@khu.ac.kr (E.-K.M.); 3Department of Medical Research Center for Bioreaction to Reactive Oxygen Species and Biomedical Science Institute, School of Medicine, Graduate School, Kyung Hee University, Seoul 02447, Korea

**Keywords:** *Toxoplasma gondii*, virus-like particle, CpG-ODN, antibody response, cytokine

## Abstract

Passive immunity can provide immediate protection against infectious pathogens. To date, only a few studies have investigated the effect of passive immunization against *Toxoplasma gondii*, and the use of immune sera acquired from VLP-vaccinated mice for passive immunity assessment remains unreported. In this study, immune sera were produced by a single immunization with virus-like particles (VLPs) expressing the inner membrane complex (IMC), rhoptry protein 18 (ROP18), and microneme protein 8 (MIC8) of *Toxoplasma gondii*, with or without a CpG-ODN adjuvant. The passive immunization of immune sera conferred protection in mice, as indicated by their potent parasite-specific antibody response, lessened brain cyst counts, lower bodyweight loss, and enhanced survival. In order to confirm that the immune sera of the VLP-immunized mice were truly protective, the antibody responses and other immunological parameters were measured in the VLP-immunized mice. We found that VLP immunization induced higher levels of parasite-specific IgG, IgG subclass, and IgM antibody responses in the sera and intestines than in the controls. Enhanced Th1 and Th2-associated cytokines in the spleen, diminished brain cyst counts, and lessened body weight loss were found following *T. gondii* ME49 challenge infection. These results suggest that passive immunization with the immune sera acquired from VLP-vaccinated mice can confer adequate protection against *T. gondii* infection.

## 1. Introduction

*Toxoplasma gondii* is a notorious intracellular protozoan parasite which infects a wide range of warm-blooded vertebrates such as cats, sheep, mice, and humans. As the causative agent of toxoplasmosis, *T. gondii* infections in healthy individuals are often asymptomatic, whereas pregnant women and acquired immune deficiency syndrome (AIDS) patients can experience severe life-threatening symptoms such as miscarriage, retinitis, and sight loss upon infection [1,2]. The major routes of human *T. gondii* infection are the ingestion of uncooked meat containing *T. gondii* cysts or oocyst-contaminated water [3]. It is estimated that one-third of the world’s population has been infected with *T. gondii* [4,5]. As of this moment, vaccination remains the most effective method for controlling infectious diseases [6]. Toxovax, albeit limited to veterinary usage, is the only commercially available *T. gondii* vaccine. For this reason, it is imperative that an additional vaccine is developed for clinical purposes [7].

To date, multiple studies have demonstrated the efficacy of a single immunization for various diseases. Immunizing mice with attenuated *Brucella melitensis* encapsulated in alginate microspheres conferred protection against challenge infection with the virulent wild type [8]. A chimeric vaccine engineered using the structural components of the yellow fever virus and the Zika virus fully protected mice against the yellow fever virus challenge following a single immunization [9]. Adjuvants are commonly incorporated into the vaccination formulae in order to bolster the protective efficacy. Evidently, the protective efficacy of a single dose of adenoviral-vectored malaria vaccine was improved through adjuvant use [10]. Synthetic oligodeoxynucleotides containing unmethylated CpG motifs (CpG ODN) are an effective adjuvant of interest with numerous studies documenting their capabilities for enhancing vaccine efficacies through the activation of both innate and acquired immunities in humans and other animals [11,12,13,14,15]. In this regard, a single immunization with the CpG-adjuvanted human papillomavirus vaccine demonstrated a prophylactic effect in mice by reducing tumor volume and emergence [16].

Studies reporting the protective efficacies of a single immunization with the *T. gondii* vaccine are extremely limited. A profound reduction of the *T. gondii* Fukaya strain was observed in mice immunized with a DNA vaccine expressing *T. gondii* heat shock protein 70 [17]. A single immunization with microparticles encompassing the chimeric surface antigen protein (rSAG1/2) protected mice against a lethal challenge with the virulent *T. gondii* RH strain [18]. To date, VLP vaccine efficacies against *T. gondii* infections were mostly evaluated following a prime-boost immunization regimen [19,20,21]. Recently, in our previous study, the protective efficacies of CpG-ODN adjuvanted *T. gondii* VLPs from mice receiving a differing number of immunizations were assessed [22,23]. Although limited information has been presented for a single immunization, the results of our previous study revealed that VLP-immunization induces potent levels of parasite-specific antibody responses, which signified its potential application in passive immunization studies.

Passive immunization refers to the process of providing antibodies to protect against infection. Their role in protection is well-established, especially against respiratory viruses and enteric pathogens [24,25]. To date, only a few studies have investigated the effect of passive immunity against toxoplasmosis, with conflicting findings. While several groups reported the protective role of passive immunization against toxoplasmosis [26,27], others have demonstrated that passive antibody immunization merely suppresses the humoral immune response, and does not prolong the survival of challenge-infected animals [28,29,30]. The underlying cause of this discrepancy can be attributed to the method of active immunization and the route selected for passive immunization. The active immunization to generate antibodies for the aforementioned passive immunity studies was performed using either killed parasites or experimental infection with *T. gondii*, and the passive immunizations were predominantly performed through the intraperitoneal route [29,30]. Antigen immunogenicity can be affected during the inactivation process, which results in weak immune response induction. As such, the antibody titers induced by inactivated vaccines are weak and tend to fade over time, thus requiring multiple booster doses [31]. The major drawback of the intraperitoneal antibody administration method is the fact that inoculated substances must undergo hepatic metabolism prior to entering the circulatory system, which incurs inoculum loss and possibly leads to erroneous results [32]. Intravenous injection is considered to be the most efficient method of substance delivery because the need for solute absorption is virtually non-existent [32]. Therefore, addressing these issues through the intravenous administration of highly immunogenic VLP-induced antibodies for passive immunization could elucidate their protective role in toxoplasmosis. In our current study, we evaluated the adjuvanted VLP vaccine-induced antibody responses (IgG, IgG1, IgG2a, IgG2b, IgM) and attempted to accurately assess the protection conferred by passively immunizing naïve mice with the sera of VLP-immunized mice through the intravenous route.

## 2. Materials and Methods

### 2.1. Ethics Statement

All of the animal experiments in this study were carried out under the guidelines set out by Kyung Hee University IACUC. The experimental protocols were approved by the Animal Ethics Committee of the Kyung Hee University (permit number: KHUASP(SE)-18-050). The immunization and blood collection were performed under mild anesthesia, which was induced and maintained with ketamine hydrochloride and xylazine. All efforts were made to minimize the number of animals used in the experiment as well as their suffering.

### 2.2. Animals, Parasites, Cells and Antibodies

The *Toxoplasma gondii* RH and ME49 strains were maintained in seven-week-old female BALB/c mice purchased from NARA Biotech (Seoul, Korea), as previously described [22]. The *Spodoptera frugiperda* Sf9 insect cells (ATCC, CRL-1711) were cultured in serum-free SF900-II medium (Invitrogen, Carlsbad, CA, USA), and were subsequently used to generate recombinant baculovirus (rBV) and virus-like particles (VLPs). Polyclonal *T. gondii* antibodies isolated from the blood of *T. gondii* ME49-infected mice were used to determine the *T. gondii*-specific antibody responses. Horseradish peroxidase (HRP)-conjugated goat anti-mouse IgG, IgG1, IgG2a, IgG2b, and IgM were purchased from Southern Biotech (Birmingham, AL, USA).

### 2.3. Passive Immunization with the Sera of VLP-Immunized Mice

In order to assess the protection conferred by passive immunization with the sera of VLP-immunized mice, seven-week-old female BALB/c mice (*n* = 6 per group) were orally infected with a lethal dose of *T. gondii* ME49 (450 cysts). After 16 h, 100 µL of the pooled sera from TG 146 VLPs or TG 146 VLPs + CpG-immunized mice were injected into the tail veins of ME49-infected mice. The control group mice received 100 µL of PBS (unimmunized).

### 2.4. Parasite-Specific IgG Antibody Responses in Mice Passively Immunized with VLPs

*T. gondii* RH antigens were prepared by sonicating tachyzoites in PBS, as described previously [22]. The *T. gondii* RH antigen-specific IgG antibody responses were determined by ELISA, as described [21]. Briefly, *T. gondii* RH antigens were coated at a concentration of 4 µg/mL in 96-well plates. The sera of passively-immunized mice were collected 30 days post-immunization, and 100 µL of the pooled sera (1:100 in PBS) were added to the respective wells, and then the plates were subsequently incubated at 37 °C for 1 h. Afterward, 100 µL of HRP-conjugated goat anti-mouse IgG antibodies (1:2000 in PBST) were added and incubated at 37 °C for 1 h. The optical density (OD) at 450 nm was read using an ELISA reader (EZ Read 400, Biochrom, Cambridge, UK).

### 2.5. Protection in Mice Passively Immunized with Sera from VLPs Immunization

In order to determine the protection offered by passive immunity, naïve mice were infected with a lethal dose of *T. gondii* ME49, and received the immune sera from VLP immunization as described above. The mice were monitored daily for 30 days in order to evaluate their changes in bodyweight and survival. At 30 days post-infection (dpi), the mouse brain tissues were isolated for cyst counting, as previously described [22]. Mice that lost 20% of initial bodyweight were humanely euthanized, as described [21,23].

### 2.6. Generation of Recombinant Baculovirus and Virus-Like Particles

Multi-antigenic TG146 VLPs were generated as previous described [20], and these were inoculated into mice in order to raise the antibodies required for passive immunization. Briefly, recombinant baculoviruses (rBVs) expressing the *T. gondii* inner membrane complex (TG1), *T. gondii* rhoptry protein 18 (TG4), *T. gondii* microneme protein 8 (TG6) and influenza M1 were produced. Sf9 cells were simultaneously infected with these rBVs for TG146 VLP production, and were incubated at 27 °C for 3 days. The TG146 VLPs were purified and subsequently quantified by BCA protein assay.

### 2.7. VLPs Immunization and Challenge

In order to prepare the immune sera for passive immunization, seven-week-old female BALB/c mice were intranasally immunized singularly with either unadjuvanted or CpG ODN-adjuvanted TG146 VLP vaccines. The immunized mice were challenge-infected orally with a lethal dose of 450 ME49 cysts 30 days after immunization. Mice that lost 20% of their initial bodyweight were considered dead and were humanely euthanized as described [21,23].

### 2.8. Mouse Sample Collection

Sera collected at weeks 1 and 4 after immunization, and week 1 post-challenge infection were used to determine the *T. gondii* specific-antibody response. At 30 dpi, three mice from each group were sacrificed, and their brain, spleen, and duodenum were collected. Isolated samples were individually processed, as previously described [33,34,35,36]. The remaining mice from each group were monitored until 30 dpi in order to measure their body weight changes and survival rates.

### 2.9. Antibody Responses in the Sera and Intestines

*T. gondii* RH antigen-specific IgG, IgG1, IgG2a, IgG2b, and IgM antibody responses were determined by ELISA, as previously described [21]. As primary antibodies, 100 µL of the diluted sera (1:100 in PBS), or undiluted brain and duodenum samples were added to respective wells, and then the plates were subsequently incubated at 37 °C for 1 h. Afterward, 100 µL of HRP-conjugated goat anti-mouse secondary antibodies (IgG, IgG1, IgG2a, IgG2b, and IgM diluted 1:2000 in PBST) were added and incubated at 37 °C for 1 h. The optical density (OD) at 450 nm was read using an ELISA reader (EZ Read 400, Biochrom, Cambridge, UK).

### 2.10. Analysis of Cytokines in the Brain and Spleen

Supernatants isolated from the brain tissue homogenates and spleen tissue were used to determine the concentrations of cytokine tumor necrosis factor-alpha (TNF-α), interferon-gamma (IFN-γ), and IL-4 and interleukin-10 (IL-10). BD OptEIA TNF-α, IFN-γ and IL-10 ELISA kits (BD Biosciences, San Jose, CA, USA) were used, and the cytokine concentrations were determined following the manufacturer’s instructions.

### 2.11. Collection of T. gondii ME49 Cysts in the Brain

The cysts were purified prior to counting using the method previously described [20]. After removing the supernatant from the brain tissue homogenates, pelleted tissues were used to count the *T. gondii* ME49 cysts. The sedimented tissues were resuspended in 40% Percoll solution and centrifuged at 12,100 rpm for 20 min. The layer containing the *T. gondii* ME49 cysts was carefully collected and centrifuged at 6000 rpm for 20 min with repeated washing in PBS. The pellets were resuspended in PBS, and 5 µL of the collected cysts were placed on a slide glass for counting under the microscope (Leica DMi8, Leica, Wetzlar, Germany). A total of 5 different fields were viewed for each sample, and the cyst counts were normalized to the total volume.

### 2.12. Statistical Analysis

The statistical analysis was performed using GraphPad Prism version 5 (San Diego, CA, USA). The data were analyzed using one-way ANOVA with Tukey’s post hoc test, or 2-way ANOVA with Bonferroni’s test. The statistical significance was denoted using asterisks (* *p* < 0.05, ** *p* < 0.01, *** *p* < 0.001).

## 3. Results

### 3.1. Experimental Schedule for Passive Immunization with Immune Sera from Single VLPs Immunization

As seen in Figure 1A, naïve mice were infected with *T. gondii* (ME49), and immune sera were intravenously administered after 16 h. In order to determine protection, the bodyweight and survival rate of the mice were monitored, and cyst counts were determined at 30 dpi. As seen in Figure 1B, the immune sera for passive immunization were obtained from a single VLP immunization. The parasite-specific antibody responses in sera and intestines, Th1- or Th2-like cytokine responses, and protection were determined as scheduled.

### 3.2. Passive Immunization with Sera from Mice Immunized with VLPs Showed Higher Levels of Parasite-Specific IgG Antibody Response and a Significant Reduction of Cyst Counts in the Brain

The parasite-specific IgG antibody responses and cyst counts in the brain were determined at 30 dpi. As indicated in Figure 2A, the parasite-specific IgG antibody responses were found to be higher in mice passively immunized with the immune sera compared to the unimmunized controls. Of the two immune sera used for the passive immunization, the mice receiving the immune sera of CpG-ODN adjuvanted VLPs elicited greater IgG antibody responses. The brain cyst counts were inversely proportional to the parasite-specific IgG responses. As shown in Figure 2B, the lowest cyst counts were observed from mice passively immunized with the sera of adjuvanted VLPs. These results indicate that the sera from immunized mice are capable of reducing the parasite loads in the brain.

### 3.3. Passive Immunization with the Sera of VLP-Immunized Mice Provided Protection against T. gondii Infection

In order to evaluate the protective efficacy of passive immunization, naïve mice initially infected with a lethal dose of *T. gondii* ME49 were subsequently inoculated with immune sera from VLP-vaccinated mice 16 h post-infection. Their bodyweight change (Figure 3A), and survival rate (Figure 3B) were determined at day 30 post-challenge infection. Passive immunization using the sera of adjuvanted and unadjuvanted VLP-immunized mice resulted in 14.8% and 18.4% bodyweight loss, respectively (Figure 3A). Passive immunization with the immune sera ensured the survival of the immunized mice. While all of the control mice died within 30 days, all of the mice receiving adjuvanted VLPs’ immune sera survived, whereas only 75% of the mice immunized with unadjuvanted VLPs’ sera survived (Figure 3B). These results indicate that the sera from the immunized mice provided protection against *T. gondii* ME49 infection.

### 3.4. VLPs Were Generated, and VLP Immunization Induced T. gondii-Specific IgG, IgG Subclass, and IgM Antibody Responses in the Sera

In order to prepare the immune sera for passive immunization, VLPs containing the *T. gondii* inner membrane complex (TG1), *T. gondii* rhoptry protein 18 (TG4) and *T. gondii* microneme protein 8 (TG6) were produced and characterized as described previously [22]. The mice were immunized with VLPs, and parasite-specific IgG, IgG1, IgG2a, IgG2b and IgM antibody responses at weeks 1 and 4 post-immunization and post-challenge were determined as scheduled. Adjuvanted VLP immunization showed higher levels of IgG, IgG1, and IgG2a antibody responses compared to unadjuvanted VLPs at week 4 (Figure 4A–C). Interestingly, compared to the unadjuvanted VLP group; a significantly higher IgG2a antibody response from adjuvanted VLP immunization was observed as early as week 1 post-immunization (Figure 4C). While noticeable differences in IgG2b levels were noted at 4 weeks post-immunization between the two immunization groups, a significant difference was only observed post-challenge infection (Figure 4D). These results indicated that CpG-ODN immunization induced an IgG2a-dominant antibody response. As seen in Table 1, adjuvanted VLP immunization showed higher levels of IgG2a antibody response compared to unadjuvanted VLP alone, implying that adjuvanted VLP immunization induced a Th1-dominant immune response. IgM is the first antibody to appear following the initial exposure to an antigen. As seen in Figure 4E, higher levels of parasite-specific IgM antibody responses in sera were detected in adjuvanted VLPs compared to those from unadjuvanted VLPs. Interestingly, higher levels of parasite-specific IgM antibody responses from both adjuvanted and unadjuvanted VLPs were elicited at week 1 compared to those at week 4, indicating that IgM antibody responses were induced at the early stage of VLP vaccine immunization. These results indicate that VLP immunized immune sera contain parasite-specific IgG antibody responses that contribute to protection against *T. gondii* (ME49) infection.

### 3.5. VLPs Immunizations Showed Parasite-Specific IgG, IgG Subclass, and IgM Antibody Responses in the Intestines

The immune responses in the intestinal mucosa can confer protection against infectious pathogens. In this study, the mice were intranasally immunized with VLPs with or without CpG-ODN adjuvant, and the parasite-specific IgG, IgG subclass, and IgM antibody responses from the intestines were assessed. Adjuvanted VLP immunization elicited higher levels of intestinal IgG, IgG subclass, and IgM antibody responses compared to unadjuvanted VLP alone (Figure 5A–E). Consistent with the serum antibody responses, adjuvanted VLP vaccination induced a more robust Th1-dominant immune response compared to unadjuvanted VLP alone (Table 1). As seen in Table 1, the Th1:Th2 index from adjuvanted VLPs and VLPs alone were statistically analyzed. The Th1:Th2 index from adjuvanted VLPs was significantly higher compared to VLPs alone.

### 3.6. VLPs Immunization Induced Cellular Immunity

The cytokines IFN-γ, IL-4 and IL-10 were measured from the splenocytes in order to determine the cellular immunity. As shown in Figure 6, the adjuvanted VLPs showed a significantly higher level of Th1-like cytokine (IFN-γ) response compared to unadjuvanted VLPs (Figure 6A). The adjuvanted VLPs showed Th2-like cytokine (IL-4, IL-10) responses, although their expressions were relatively lower than the Th1-like cytokine IFN-γ. These results indicated that adjuvanted VLPs induced Th1 cytokine dominant immune responses, which is consistent with the Th1-dominant immune responses observed from the sera and intestines described above.

### 3.7. VLPs Immunization Induced Protective Efficacy

VLP immunizations with or without adjuvant CpG showed significantly lower levels of inflammatory cytokine TNF-α in the brain upon challenge infection compared to the unimmunized naïve control (Naïve+cha). Importantly, the adjuvanted VLPs showed significantly lower levels of TNF-α response compared to the unadjuvanted VLPs (Figure 7A). *T. gondii* ME49 cysts were counted from the brains of mice under the microscope in order to compare the protective efficacies of the unadjuvanted and adjuvanted VLP vaccines. As seen in Figure 5B, the adjuvanted VLPs showed much lower cyst counts in the brain compared to unadjuvanted VLPs, indicating that the CpG ODN adjuvant contributed to the protection. Additionally, immunization with adjuvanted VLPs incurred lesser bodyweight loss than the unadjuvanted VLPs (Figure 7C).

## 4. Discussion

In this study, we evaluated the passive immunity induced by the immune sera of VLP-immunized mice. We found that the passive immunization of *T. gondii*-specific antibodies produced by VLP immunization, irrespective of adjuvant usage, functioned as an effective prophylaxis tool against *T. gondii* infection in mice. As expected, the passively immunized mice demonstrated higher parasite-specific IgG responses than the unimmunized controls at 30 dpi. The antibody responses in the passively immunized mice were proportional to the level of protection conferred upon challenge infection. Evidently, the highest parasite burden reduction was observed from the mice passively immunized with the sera of adjuvanted VLP-immunized mice. This was further exemplified by their 100% survival following challenge infection with minimal bodyweight loss. Unadjuvanted VLP-immunized mice sera also conferred protection in the passively immunized mice, albeit to a lesser extent than their adjuvanted counterpart.

Neutralizing antibodies in passive immune therapy can be used as an effective therapeutic alternative for many viral diseases, including the influenza virus, measles virus, Ebola virus, respiratory syncytial virus (RSV), and so on [37]. In the case of *T. gondii*, conflicting results were reported from multiple studies investigating the role of passive immunization. Earlier studies by Foster and McCulloch [30] revealed that *T. gondii* persistence in tissues is still possible even with high anti-toxoplasma antibody titers, and the heterologous transfer of passive immunity from guinea pigs to mice does not significantly prolong the survival of the recipient animals. The possibility of passively administered antibodies suppressing the humoral response in the recipient animals was also reported [28,29]. However, recent studies have demonstrated that passive immune therapy can confer partial protection in various animal models. Passively immunizing IgM-ablated mice with the sera of chronic *T. gondii*-infected mice enabled 50% survival [38]. Another study reported that antibodies raised against toxoplasmosis inhibited the number of tachyzoites resembling a rosette conformation, but their passive administration had no effect on the replication rate [39]. Administering the sera of *T. gondii* RH-infected guinea pigs into recipients conferred partial protection [26]. The intraperitoneal administration of anti-toxoplasma serum in rabbits conferred protection against lethal challenge infection with the *T. gondii* Alt strain, as indicated by a 55% brain cyst reduction and a 89% survival rate [27]. Consistent with these recent findings reported above, our study confirmed that passive immunization using anti-*T. gondii* antibodies can confer protection against *T. gondii* infection. This is further supported by our previous study using VLPs expressing ROP18 and MIC8 antigens. Both ROP18 and MIC8 VLP-induced antibodies were capable of neutralizing the *T. gondii* GT1 strain to a small extent, but parasite-neutralizing activity exceeding 50% was observed when the immune sera of ROP18+MIC8 VLP-immunized mice reacted with the *T. gondii* GT1 [40].

*T. gondii* can be subdivided into three major genotypes, which are classified as types I, II, and III. Type I strains are generally virulent and lethal, whereas type II strains are less virulent than type I. Type III strains, such as C56, were reported to be rarely associated with the disease [41,42]. RH and GT1, both belonging to the type I clonal lineage, were reported to share a high degree of sequence similarity [43]. Based on this rationale, immunity developed against GT1 strain could also be applied to RH strain as well. The potential for heterologous protection can also be exemplified using our current and previous studies. Previously, we demonstrated that VLPs based on ROP18 and MIC8 conferred protection against the GT1 strain and exhibited strong parasite neutralizing activity [40]. Sera acquired from mice immunized with the TG146 VLP vaccine, which is identical to the VLP vaccine used in the current study, were capable of neutralizing as much as 60% of the *T. gondii* GT1 strain [20]. Here, we showed that passively immunizing the sera raised against this TG146 VLP conferred protection against the type II ME49 strain. Combined, our results imply that IMC, ROP18, and MIC8-expressing VLP immunization induces the production of broadly neutralizing antibodies that can effectively restrict the growth of type I and II *T. gondii* strains.

Next, we assessed the protective immunity in mice immunized with VLPs encoding the IMC, ROP18, and MIC8 of *T. gondii.* Consistent with the protection observed from passively immunized mice, significantly higher levels of parasite-specific IgG, IgG subclasses, and IgM antibodies were observed from the sera and intestinal mucosa of adjuvanted VLP-immunized mice than the unadjuvanted VLP group. CpG ODN stimulates Toll-like receptor 9 (TLR9) in lymphocytes, and induces the production of cytokines such as IFN-γ, IL-6, IL-10, and TNF-α [44]. CpG ODN-adjuvanted vaccines are under development for the prevention of cancer, allergies, and various infectious diseases [45]. To date, only a limited number of studies have investigated the protective efficacies of *T. gondii* vaccines adjuvanted with CpG. Recently, it was reported that immunization with *T. gondii* lysate antigen and CpG induces *T. gondii*-specific antibody responses and IFN-γ production in the spleen and MLN of mice [46]. Consistent with these reports, our study indicated that CpG-adjuvanted VLP immunization induced higher levels of parasite-specific IgG2a antibody responses in the sera and production of cytokines IFN-γ and IL-10 in comparison to the unadjuvanted VLPs. On a similar note, the enhanced IgG1 responses observed from adjuvanted VLPs contributed to a marginal increase in the Th2 cytokine IL-4 expression. Elevated IgG2a and IgG2b serum and intestinal antibody responses, paired with the heightened IgA response reported in our previous study via adjuvanted VLPs [22], may have contributed to better protection against *T. gondii* than the unadjuvanted VLPs. Interestingly, the IFN-γ production in adjuvanted VLP-immunized mice far exceeded the concentrations of IL-10, while such a phenomenon was not observed from unadjuvanted VLP-immunized mice. The striking difference in IFN-γ production between the two immunization groups can be attributed to the presence of CpG ODN adjuvants, thereby implying their contribution and necessity for mounting a potent Th1-dominant immune response.

IgM is the first antibody secreted by the adaptive immune system in response to a foreign antigen. However, it is widely neglected when evaluating vaccine-induced immunity, as IgG and IgA antibodies are the main targets for immune response assessment. IgM molecules, characterized by their ten antigen-binding sites that contribute to their high antigen avidity, can neutralize pathogens and protect mucosal surfaces [47]. In this study, VLP immunizations induced parasite-specific IgM in sera and intestines, which may contribute to the protection against *T. gondii* challenge infection. Furthermore, serum IgM antibodies have been shown to contribute to high levels of opsonophagocytic activities in a single dose immunization study with pneumococcal conjugate vaccine [48]. The role of opsonophagocytic activity contributing to the protection against *T. gondii* infection needs further study.

In conclusion, our results demonstrated that a single immunization with the TG146 VLPs induced the production of neutralizing antibodies, conferring protection against *T. gondii* in mice. Passively immunizing mice with the sera of VLP-immunized mice revealed that the sera were protective irrespective of the adjuvant inclusion. Formulating CpG adjuvants with TG146 VLPs enhances the parasite-specific IgG, IgG subclass, and IgM antibody responses, thereby indicating their potential as a passive immune therapeutic option.

## Figures and Tables

**Figure 1 vaccines-09-00425-f001:**
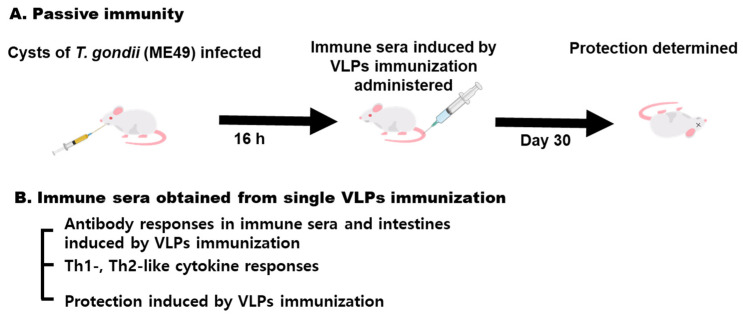
Experimental schedule for passive immunization. (**A**) Naïve mice were initially infected with *T. gondii* (ME49), and the immune sera of VLP-immunized mice were subsequently administered. Protection was determined at 30 dpi. (**B**) Immune sera were obtained after a single VLP immunization from mice, and their antibody responses, Th1-, Th2-like cytokine responses, and protection were determined.

**Figure 2 vaccines-09-00425-f002:**
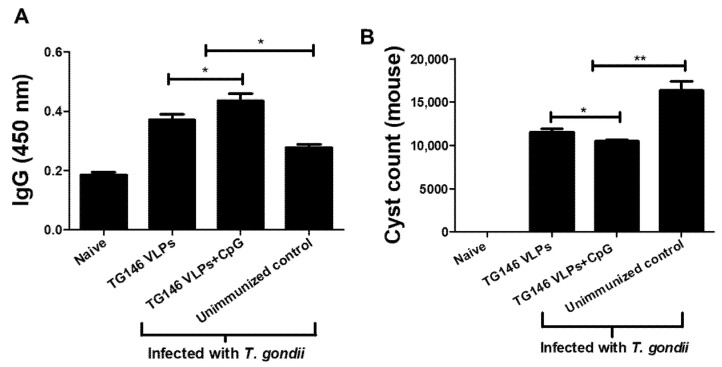
Parasite-specific IgG antibody response and cyst counts in the brain from mice which received the immune sera. The mouse sera were collected at 30 dpi, and their parasite-specific IgG antibody responses (**A**) and cyst counts were assessed (**B**). The data are expressed as mean ± SD, and statistical significance is denoted using asterisks (* *p* < 0.05, ** *p* < 0.01).

**Figure 3 vaccines-09-00425-f003:**
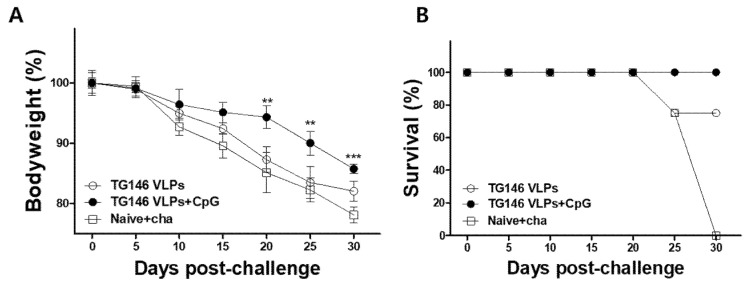
Passive immunization using the sera from VLP-immunized mice conferred protection against parasite infection. Naïve mice were infected with a lethal dose of *T. gondii* (ME49), and immune sera from VLP-vaccinated mice were injected into the tail vein. Their body weight change (**A**), and survival rate (**B**) were determined. The sera from adjuvanted VLPs showed better protection compared to those from unadjuvanted VLP immunization. The data are expressed as mean ± SD, and statistical significance is denoted using asterisks (** *p* < 0.01, *** *p* < 0.001).

**Figure 4 vaccines-09-00425-f004:**
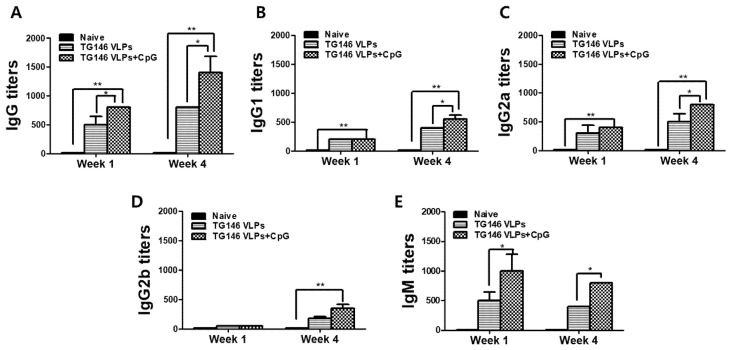
IgG, IgG subclasses, and IgM antibody responses in sera. Sera collected from mice post-immunization were used to assess the *T. gondii*-specific antibody responses. The levels of IgG (**A**), IgG1 (**B**), IgG2a (**C**), IgG2b (**D**), and IgM (**E**) antibody responses in the CpG-adjuvanted TG146 VLPs were higher than those from the TG146 VLPs alone. The data are expressed as mean ± SD, and statistical significance is denoted using asterisks (* *p* < 0.05, ** *p* < 0.01, *** *p* < 0.001).

**Figure 5 vaccines-09-00425-f005:**
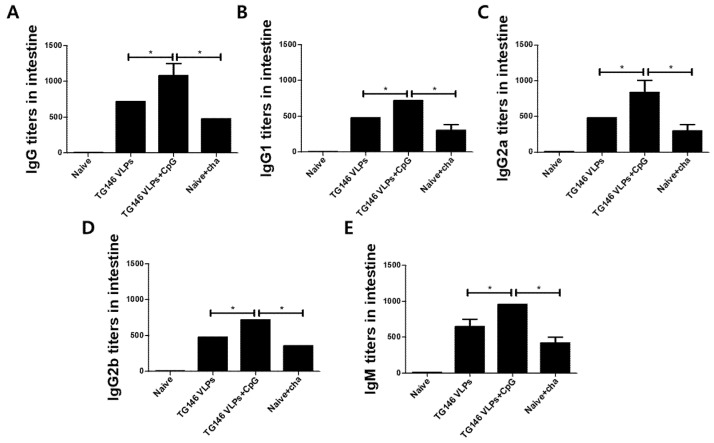
*T. gondii*-specific antibody responses in intestinal samples. The *T. gondii*-specific antibody responses were assessed from the intestinal samples by ELISA. The IgG (**A**), IgG subclasses (**B**–**D**), and IgM (**E**) antibody responses from the duodenum were measured. Adjuvanted VLP immunizations showed higher levels of parasite-specific antibody responses compared to the unadjuvanted VLPs. The data are expressed as the mean ± SD, and statistical significance is denoted using asterisks (* *p* < 0.05, ** *p* < 0.01).

**Figure 6 vaccines-09-00425-f006:**
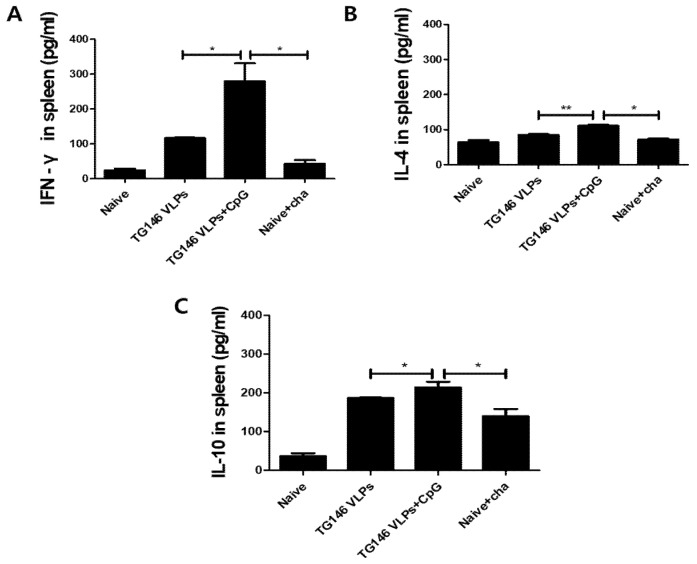
Cellular immunity induction. Mice were immunized with VLPs with or without adjuvant and challenge-infected with *T. gondii* (ME49). Th1-like cytokine IFN-γ (**A**) and Th2-like cytokines IL-4 (**B**) and IL-10 (**C**) in the spleen were determined to assess cellular immunity. As seen in Figure 6, adjuvanted VLPs immunization induced higher levels of cytokines IFN-γ, IL-4 and IL-10 compared to unadjuvanted VLPs. Data are expressed as mean ± SD and statistical significance is denoted using asterisks (* *p* < 0.05, ** *p* < 0.01).

**Figure 7 vaccines-09-00425-f007:**
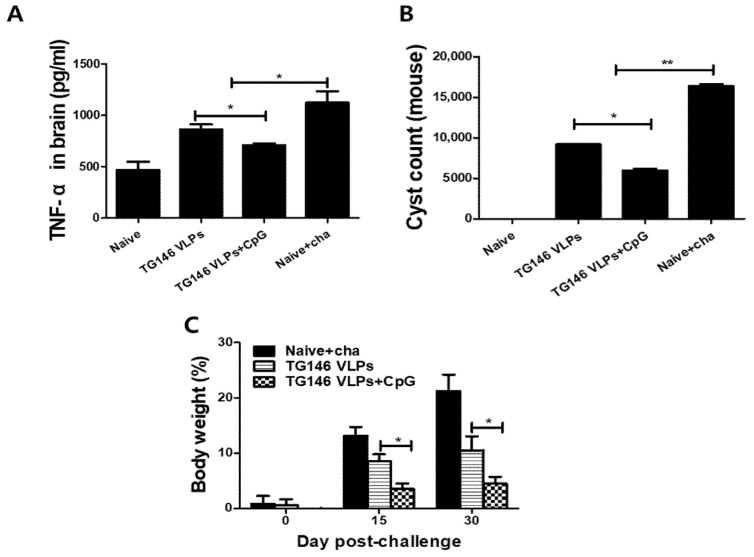
Protective efficacy induced by VLP immunizations. The VLP-immunized mice were challenge-infected with *T. gondii* (ME49), and the mice were sacrificed 30 dpi for brain tissue collection. The adjuvanted VLP immunization showed significantly lower levels of inflammatory cytokine TNF-α (**A**) compared to unadjuvanted VLPs alone. The cysts in the brain were counted (**B**) and the bodyweight loss was determined (**C**). The data are expressed as the mean ± SD, and statistical significance is denoted using asterisks (* *p* < 0.05, ** *p* < 0.01).

**Table 1 vaccines-09-00425-t001:** Th1:Th2 index for VLPs alone, and for adjuvanted VLPs.

Groups (*n* = 6)	Serum		Intestine	
	TG146	TG146 + CpG	TG146	TG146 + CpG
IgG1 (Th2)	0.13 ± 0.002	0.16 ± 0.004	0.39 ± 0.008	0.45 ± 0.005
IgG2a (Th1)	0.18 ± 0.001	0.29 ± 0.003	0.55 ± 0.007	0.66 ± 0.006
IgG2b (Th1)	0.11 ± 0.006	0.14 ± 0.004	0.33 ± 0.002	0.45 ± 0.003
Th1:Th2 Index	1.11 ± 0.004	1.32 * ± 0.003	1.12 ± 0.003	1.23 * ± 0.004

The sera and intestines were collected at day 30 post-challenge infection, and the IgG1 and IgG2a antibody responses in both the sera and intestines were determined. The values are optical density readings at 100-fold serum dilution. Index <1 = Th2 polarization. Index >1 = Th1 polarization. Th1 (IgG2a and IgG2b) or Th2 (IgG1) polarization were determined by Th1:Th2 index calculation ([IgG2a + IgG2b]/2)/(IgG1), as previously described [11] (* *p* < 0.05).

## Data Availability

Not applicable.

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
