# Peer review of "Passive Immunity and Antibody Response Induced by Toxoplasma gondii VLP Immunization"

_vaccines, 2021, doi:10.3390/vaccines9050425_

Round 1

Reviewer 1 Report

The development of an effective and safe vaccine, able to produce a durable immune response remains a necessity to facilitate the control of toxoplasmosis. The present manuscript evaluates the use of a virus-like particles (VLPs) expressing the inner membrane complex, ROP18, and MIC8 of Toxoplasma gondii with or without CpG-ODN administered as a single dose vaccine for protection against Toxoplasma gondii infection. The work is well designed, and the results are interesting. However, I have several major concerns which I describe below.

Major

  • I do not see big differences compared to their previous publication (Evaluation of CpG-ODN-Adjuvanted Toxoplasma gondii Virus-Like Particle Vaccine upon One, Two, and Three Immunizations. Pharmaceutics 2020, 12, 989). It does not seem to me that the current manuscript adds much to what has already been published by the authors previously. Indeed, the conclusion of the current manuscript that formulating CpG adjuvants with TG146 VLPs enhances the protective efficacy against gondii ME49 challenge infection in mice is similar to the previous paper. Herein the authors emphasize the use of a single-dose vaccine. However, in their previous publication they already demonstrated that a single immunization protected (100% survival) from the challenge. It would be interesting to study if immunization with a single dose induces a long-term memory response as good as with 2 or 3 doses by challenging the mice at different time points.
  • There is something that I do not understand. According to Figure 6, 30 days after infection all non-vaccinated mice die. However, in Figure 5 and 6 cyst count and body weight are shown at 30 dpi. Were those parameters evaluated in dead mice?
  • Also, and in the same line, according to Materials and Methods at 30 days post-infection, three mice from each group were sacrificed and brain and spleen were collected to measure cytokines in homogenate supernatants. However, the non-vaccinated and challenged mice are all dead.
  • Duodenum was collected at 30 dpi. Are the humoral response results shown in Figure 3 from those samples?
  • In Material and Methods, line 118, authors state that brain tissues collected from mice 30 dpi were homogenized with a syringe in 400 μL PBS to measure antibodies in supernatant. No graph showing those results are included in the manuscript.
  • In Table 1: For Th1/Th2 index it is not clear how the statistic was calculated. No standard deviation is shown. Which is the sample size of each experimental group? What groups are compared to get the statistical significance (p)?
  • Figure 3. Authors should also compare to the naive+challenge group.
  • Figure 4. As for Figure 3 Authors should also compare to the naive+challenge group.
  • Regarding Figure 4: The classification of IL-6 as a TH2-like cytokine called my attention. I wouldn´t say at all that IL-6 is a TH2-like cytokine. IL-6 is a pleiotropic cytokine, and many biological activities are affected by its production. Although it was described many years ago (by 2000) that IL-6 stimulates CD4 T cells to secrete IL-4 and direct the response to Th2, presently it is most known 1) as a proinflammatory cytokine; 2) as an IgG stimulating cytokine by regulating the expression of IL-21 and 3) in combination with TGF-β, IL-6 induces the differentiation of naïve CD4 into Th17 cells.

To evaluate TH2 cytokines, authors should measure IL-4, IL-5 or IL-13

Nevertheless, I recognize the importance of measuring IL-6 in that it is a necessary cytokine to prevent the progression of many infectious diseases.

Minor

  • The Journal name is missing in reference 23.
  • Table 1: For Th1/Th2 index, at what time point was calculated?

Author Response

Response to Reviewers’ Comments

Reviewer 1

The development of an effective and safe vaccine, able to produce a durable immune response remains a necessity to facilitate the control of toxoplasmosis. The present manuscript evaluates the use of a virus-like particles (VLPs) expressing the inner membrane complex, ROP18, and MIC8 of Toxoplasma gondii with or without CpG-ODN administered as a single dose vaccine for protection against Toxoplasma gondii infection. The work is well designed, and the results are interesting. However, I have several major concerns which I describe below.

Major

  • I do not see big differences compared to their previous publication (Evaluation of CpG-ODN-Adjuvanted Toxoplasma gondii Virus-Like Particle Vaccine upon One, Two, and Three Immunizations. Pharmaceutics 2020, 12, 989). It does not seem to me that the current manuscript adds much to what has already been published by the authors previously. Indeed, the conclusion of the current manuscript that formulating CpG adjuvants with TG146 VLPs enhances the protective efficacy against gondiiME49 challenge infection in mice is similar to the previous paper. Herein the authors emphasize the use of a single-dose vaccine. However, in their previous publication they already demonstrated that a single immunization protected (100% survival) from the challenge.

Response: Thank you for the comment. In the paper (Pharmaceutics 2020, 12, 989), the protective immunity induced by multiple immunizations was emphasized whereas very limited information from a single immunization was provided. In the current study, we focused on assessing the protective efficacy induced via single immunization and passive immunization. Parasite-specific IgG, IgM, and IgG subclass IgG1, IgG2a, IgG2b antibody responses from sera and intestine, as well as Th1 and Th2-like cellular immunity were newly assessed. Inflammatory cytokine TNF-α in the brain was also newly determined. Importantly, protection induced by passive immunization with the sera of VLP-immunized mice against T. gondii infection was assessed.    

  • It would be interesting to study if immunization with a single dose induces a long-term memory response as good as with 2 or 3 doses by challenging the mice at different time points.

Response: Thank you for the comment. Long-term immunity study has been scheduled and we will perform the experiment soon.

  • There is something that I do not understand. According to Figure 6, 30 days after infection all non-vaccinated mice die. However, in Figure 5 and 6 cyst count and body weight are shown at 30 dpi. Were those parameters evaluated in dead mice?

Response: In the current study, mice that lost 20% in body weight were humanely euthanized as previously described (Vaccine, 2018, 36, 5692; Frontiers in Immunology, 2019, 9, 3073). This point has been newly added (line 117-118). Figure 5 illustrates the protection induced by a single immunization, with all of the unimmunized mice reaching the humane intervention point at 35 dpi (20% or greater loss of initial bodyweight). Datasets in Fig 6 represent the protection induced in mice following passive immunization with the sera of VLP-immunized mice. Similar to the Fig 5 data, unimmunized mice were humanely euthanized at 30 dpi (indicated by 20% or more bodyweight loss). In short, immunized mice were simultaneously sacrificed along with the unimmunized controls for organ sampling, which corresponds to the day when the unimmunized mice reached humane intervention points.

  • Also, and in the same line, according to Materials and Methods at 30 days post-infection, three mice from each group were sacrificed and brain and spleen were collected to measure cytokines in homogenate supernatants. However, the non-vaccinated and challenged mice are all dead.

Response: As mentioned above, at 30 days post-infection, mice that lost 20% of body weight were humanly euthanized, and the spleen and brain were collected to measure cytokine responses.

  • Duodenum was collected at 30 dpi. Are the humoral response results shown in Figure 3 from those samples?

Response: Yes, Fig 3 illustrates the humoral responses induced in the duodenum. The mice duodenum samples were collected at 30 dpi, and parasite-specific IgG, IgM, and IgG subclasses were determined.

  • In Material and Methods, line 118, authors state that brain tissues collected from mice 30 dpi were homogenized with a syringe in 400 μL PBS to measure antibodies in supernatant. No graph showing those results are included in the manuscript.

Response: It has been removed.

  • In Table 1: For Th1/Th2 index it is not clear how the statistic was calculated. No standard deviation is shown. Which is the sample size of each experimental group? What groups are compared to get the statistical significance (p)?

Response: The standard deviation and sample size are newly added in table 1 and statistical significance was compared between the TG 146 and TG 146+CPG groups.

  • Figure 3. Authors should also compare to the naive+challenge group.

Response: They have been compared.

  • Figure 4. As for Figure 3 Authors should also compare to the naive+challenge group.

Response: They have been compared.

  • Regarding Figure 4: The classification of IL-6 as a TH2-like cytokine called my attention. I wouldn´t say at all that IL-6 is a TH2-like cytokine. IL-6 is a pleiotropic cytokine, and many biological activities are affected by its production. Although it was described many years ago (by 2000) that IL-6 stimulates CD4 T cells to secrete IL-4 and direct the response to Th2, presently it is most known 1) as a proinflammatory cytokine; 2) as an IgG stimulating cytokine by regulating the expression of IL-21 and 3) in combination with TGF-β, IL-6 induces the differentiation of naïve CD4 into Th17 cells.

Response: Thanks for the comment. Although not all IL-6 is a Th2-like cytokine as commented, at least some of IL-6 is involved in Th2-like cytokine response as described (Mayer, Alice, et al. European journal of immunology 44.11 (2014): 3252-3262; Siegrist, Claire-Anne. Vaccines 5.1 (2008): 17-36).

To evaluate TH2 cytokines, authors should measure IL-4, IL-5 or IL-13

Response: Thank you for the comment. We will measure IL-4, IL-5, or IL-13 in the near future.

Nevertheless, I recognize the importance of measuring IL-6 in that it is a necessary cytokine to prevent the progression of many infectious diseases.

Response: Thank you. We agree with your opinion.

Minor

  • The Journal name is missing in reference 23.

Response: It has been added (line 402).

  • Table 1: For Th1/Th2 index, at what time point was calculated?

Response: The sera and intestines were collected on day 30 post-challenge infection and IgG1 and IgG2a antibody responses were determined. This point has been added in the text (line 339-340).

Reviewer 2 Report

The paper by King et al makes an attempt to generate a VLP-based vaccine that protects mice against T. gondii after a single immunisation. However, the fact that it is possible to protect mice to some degree after a single immunisation has little impact on humans, giving the vastly stronger immune responses mice make compared to humans. However, it seems of significant interest that antibodies alone are able to mediate protection. However, as shown in Fig 6B, the differences are minimal. In addition, the survival curve stops at day 30; all the immunised mice probably also die within the next 2 days judged from their weight loss. In addition, Fig 6A should be labeled as in Figure 5B. Otherwise it is very confusing. Thus, to merit publication, this point should be worked up much better. 

Minar points:

Line 43: Flu vaccines are not protective after a single injection. They are given yearly and heavily rely on pre-existing T and B immunity. Polio also needs booster immunisations.

Section 2.3. The VLPs are very poorly introduced. Both in 2.3 and the results and/or introduction section

Fig 2 and 3 need antibody titrations. Not single concentrations.

Fig 3 Can you exclude antibodies from blood contamination?

Fig 5C is wrongly labelled

Author Response

Reviewer 2

The paper by King et al makes an attempt to generate a VLP-based vaccine that protects mice against T. gondii after a single immunization. However, the fact that it is possible to protect mice to some degree after a single immunization has little impact on humans, giving the vastly stronger immune responses mice make compared to humans. However, it seems of significant interest that antibodies alone are able to mediate protection. However, as shown in Fig 6B, the differences are minimal. In addition, the survival curve stops at day 30; all the immunized mice probably also die within the next 2 days judged from their weight loss. In addition, Fig 6A should be labeled as in Figure 5B. Otherwise it is very confusing. Thus, to merit publication, this point should be worked up much better. 

Response: Fig. 5 is the protection data induced by a single immunization with the VLPs, whereas Fig. 6 is the protection data induced by passive immunization using the sera from VLP-immunized mice. They are completely different experiments. Actually, body weight loss recovery was observed from all of the immunized mice and their survival ensued. Fig. 6 has been labeled as in Fig. 5B.

Minar points:

Line 43: Flu vaccines are not protective after a single injection. They are given yearly and heavily rely on pre-existing T and B immunity. Polio also needs booster immunisations.

Response: Line 43 has been removed.    

Section 2.3. The VLPs are very poorly introduced. Both in 2.3 and the results and/or introduction section

Response: More information on the VLPs has been added in 2.3 and 3.1 (lines 104-106, 168, 170-172). More information on VLPs from our previous work has been cited.

Fig 2 and 3 need antibody titrations. Not single concentrations.

Response: The antibody responses indicated in Fig 2 and 3 are not concentrations. They are optical density (OD) values determined by ELISA, which have been extensively used by numerous research groups to assess vaccine-induced antibody responses (Journal of Virology, Apr. 2007, p. 3514–3524; JID 2011:204 (1 October); Frontiers in Immunology, 2019, 9, 3073; Pharmaceutics 2020, 12, 989; Sci Rep. 2021; 11: 4151).       

Fig 3 Can you exclude antibodies from blood contamination?

Response: The blood sampling procedure was conducted under sterile conditions. Furthermore, blood contamination does not occur during the ELISA assay process since the sera used in the study are completely devoid of RBCs (which have been removed via centrifugation).

Fig 5C is wrongly labelled

Response: It has been corrected.

Reviewer 3 Report

The authors of the ms under review: "Antibody responses induced by single immunization of adjuvanted Toxoplasma gondii VLPs vaccine" have explored the effectiveness of a nasally administered, single-dose, CpG-ODN adjuvanted T.gondii vaccine, designed using Virus Like-Particles. Using mice experiments, they have shown the effectiveness of the adjuvanted vaccine over the unadjuvanted vaccine in terms of IgM and IgG responses. However, there are some serious concerns over their interpretation of the results.

  1. A similar set of authors (with the same first and last author) have published their results on the CpG-ODN adjuvanted vaccine against T.gondii in J. Parasite Immunology, in Oct2020 (https://pubmed.ncbi.nlm.nih.gov/33058167). I find the overall design and results of both their works to be highly similar and the only difference is that the published work uses an intramuscular injection instead of a nasal one in the present study. With slight changes in the presentation of their results in both the studies, the concluding findings are the same that a CpG-ODN adjuvanted VLP vaccine is highly effective against T.gondii. Besides, the accepted article is not cited in the current ms.
  2. After reading their previous work, the current ms shall rather be focussed on the advantages of the nasal vaccine over the intra-muscular vaccine. The presented argument about the syringe-based vaccine being a dissuader of vaccination in the Human population is not sufficient enough, especially when the majority of human vaccinations with great coverage are syringe-based. Therefore, it will be imperative to the current ms that the comparative results of these two studies must be included. Similar to PMID: 24728558.
  3. A major highlight of the work is the improvement shown by the adjuvanted vaccine over the unadjuvanted one using the antibody responses. The responsive increase of IgG1, IgG2a in the intestine (adjuvanted over unadjuvanted) does not seem convincing at a p-value of <0.05. The authors shall explain this small difference in detail.

Overall, I propose that the authors should either show strong differences between their J. Parasite immunology study and current ms or restructure the complete ms as a comparison between the intra-muscular and intra-nasal T.gondii vaccines explaining the benefits in details. Thus, I propose a resubmission after the major revision of the presentation and interpretation of results.

Author Response

Reviewer 3

The authors of the ms under review: "Antibody responses induced by single immunization of adjuvanted Toxoplasma gondii VLPs vaccine" have explored the effectiveness of a nasally administered, single-dose, CpG-ODN adjuvanted T.gondii vaccine, designed using Virus Like-Particles. Using mice experiments, they have shown the effectiveness of the adjuvanted vaccine over the unadjuvanted vaccine in terms of IgM and IgG responses. However, there are some serious concerns over their interpretation of the results.

  1. A similar set of authors (with the same first and last author) have published their results on the CpG-ODN adjuvanted vaccine against T.gondii in J. Parasite Immunology, in Oct2020 (https://pubmed.ncbi.nlm.nih.gov/33058167). I find the overall design and results of both their works to be highly similar and the only difference is that the published work uses an intramuscular injection instead of a nasal one in the present study. With slight changes in the presentation of their results in both the studies, the concluding findings are the same that a CpG-ODN adjuvanted VLP vaccine is highly effective against T.gondii. Besides, the accepted article is not cited in the current ms.

Response: In our previous study (Parasite Immunology, PMID: 33058167), the emphasis was placed on the protective immune response induced via multiple immunizations. While immunization response data from single immunization were also provided, they were limited. As such, in the present study, we focused on measuring the immune responses that were not assessed in our previous publications. The present study investigated the role of immune correlates that were not addressed in the previous study. Notably, parasite-specific IgM and IgG subclasses from both the sera and intestines, Th1 and Th2-like immunity, and the inflammatory cytokine TNF-α concentrations in the blood were newly assessed. We also investigated the potential role of serum-mediated protection resulting from passive immunization, which has never been addressed in any of the T. gondii VLP vaccine studies reported to date. As suggested, the paper from Parasite Immunology has been newly cited (lines 77, 272).      

  1. After reading their previous work, the current ms shall rather be focussed on the advantages of the nasal vaccine over the intra-muscular vaccine. The presented argument about the syringe-based vaccine being a dissuader of vaccination in the Human population is not sufficient enough, especially when the majority of human vaccinations with great coverage are syringe-based. Therefore, it will be imperative to the current ms that the comparative results of these two studies must be included. Similar to PMID: 24728558.

  1. Response: As described above, in the current study, we focused on single immunization of VLPs-induced antibody responses and the protection induced by passive immunization using the sera of single VLPs-immunized mice. Previously, we have already published papers involving either intranasal vaccine (Pharmaceutics) or intramuscular vaccine (Parasite immunology). To avoid confusion, the syringe-based vaccine part in the discussion section has been removed.

  1. A major highlight of the work is the improvement shown by the adjuvanted vaccine over the unadjuvanted one using the antibody responses. The responsive increase of IgG1, IgG2a in the intestine (adjuvanted over unadjuvanted) does not seem convincing at a p-value of <0.05. The authors shall explain this small difference in detail.

Response: Th1:Th2 index from adjuvanted VLPs and VLPs alone were statistically analyzed. Th1:Th2 index from adjuvanted VLPs was significantly higher compared to VLPs alone. This point has been newly added (line 199-202).            

Overall, I propose that the authors should either show strong differences between their J. Parasite immunology study and current ms or restructure the complete ms as a comparison between the intra-muscular and intra-nasal T.gondii vaccines explaining the benefits in details. Thus, I propose a resubmission after the major revision of the presentation and interpretation of results.

Response: There is a strong difference between our previous publication (Parasite Immunology paper) and the current study. In Parasite Immunology, multiple intramuscular immunization-induced VLPs vaccine efficacy was emphasized. In the current study, single intranasal immunization-induced vaccine efficacy was emphasized, in which IgM antibody response and IgG subclass antibody responses were newly assessed. Importantly, protection induced by passive immunization with the sera of VLP-immunized mice has not been reported from our group and others. 

Round 2

Reviewer 1 Report

  • Regarding my first concern, I understand and agree with author´s response in that this manuscript extends their previous work by assessing new parameters. Still, if the conclusion of the manuscript is that formulating CpG adjuvants with TG146 320 VLPs enhances the protective efficacy of single-dose vaccines against T. gondii ME49 challenge infection in mice this manuscript is not quite original as it was already demonstrated in the previous paper (Pharmaceutics 2020, 12, 989). Analyzing whether immunization with a single dose induces a long-term memory response as good as with 2 or 3 doses by challenging the mice at different time points would reinforce the conclusion that a single dose induces protection and would make the manuscript something more novel.
  • Regarding authors response to the concern about evaluating parameters in non-vaccinated mice at 30 dpi: If unimmunized mice were humanely euthanized at 30 dpi it is not correct to make a survival chart (Fig. 6). At 30 dpi mice are still alive.
  • In Material and Methods, line 118, authors state that brain tissues collected from mice 30 dpi were homogenized with a syringe in 400 μL PBS to measure antibodies in supernatant. No graph showing those results are included in the manuscript. The authors replied that they had removed it, however it was not removed. The sentence is still in the manuscript.
  • In Figures 3 and 4, authors should also compare to the naive+challenge group. The authors replied that in this new version they experimental groups were compared to the naïve one. However, they did not. I cannot see it in the graphs
  • Regarding Figure 4 I don´t agree at all with classifying IL-6 as a Th2 cytokine. Moreover, the paper cited by the authors (European journal of immunology44 (2014): 3252-3262) states the opposite: IL‐6 secreted by dendritic cells exerted a dominant, negative influence on Th2‐cell development. To evaluate TH2 cytokines, authors should measure IL-4, IL-5 or IL-13

Author Response

Response to Reviewer 1

Regarding my first concern, I understand and agree with author´s response in that this manuscript extends their previous work by assessing new parameters. Still, if the conclusion of the manuscript is that formulating CpG adjuvants with TG146 320 VLPs enhances the protective efficacy of single-dose vaccines against T. gondii ME49 challenge infection in mice this manuscript is not quite original as it was already demonstrated in the previous paper (Pharmaceutics 2020, 12, 989). Analyzing whether immunization with a single dose induces a long-term memory response as good as with 2 or 3 doses by challenging the mice at different time points would reinforce the conclusion that a single dose induces protection and would make the manuscript something more novel.

Response: Since passive immunity against toxoplasmosis using the sera acquired from VLP-immunized animals has never been evaluated, we re-assessed the parasite-specific IgG antibody responses in mice receiving the VLP immune sera as illustrated in Figure 2A. Also, we have restructured our manuscript to highlight the importance of passive immunity induced by VLP immune sera.

Regarding authors response to the concern about evaluating parameters in non-vaccinated mice at 30 dpi: If unimmunized mice were humanely euthanized at 30 dpi it is not correct to make a survival chart (Fig. 6). At 30 dpi mice are still alive.

Response: This data has been re-located to Figure 3. As commented, we have corrected the figure to show that the survival rate of unimmunized mice (Naïve+cha group, which were humanely euthanized) is 0% at 30 dpi (Figure 3B).

  • In Material and Methods, line 118, authors state that brain tissues collected from mice 30 dpi were homogenized with a syringe in 400 μL PBS to measure antibodies in supernatant. No graph showing those results are included in the manuscript. The authors replied that they had removed it, however it was not removed. The sentence is still in the manuscript.

Response: It has been removed.

In Figures 3 and 4, authors should also compare to the naive+challenge group. The authors replied that in this new version they experimental groups were compared to the naïve one. However, they did not. I cannot see it in the graphs

Response: These data have been moved and now corresponds to Figure 5 and 6, respectively. As commented, we have included the naïve+cha data for comparison.

Regarding Figure 4 I don´t agree at all with classifying IL-6 as a Th2 cytokine. Moreover, the paper cited by the authors (European journal of immunology44 (2014): 3252-3262) states the opposite: IL‐6 secreted by dendritic cells exerted a dominant, negative influence on Th2‐cell development. To evaluate TH2 cytokines, authors should measure IL-4, IL-5 or IL-13

Response: The cytokine data has been re-located to Figure 6. The classic Th2 cytokine IL-4 level has been newly measured and included as commented (Figure 6B).

Reviewer 2 Report

The authors did essentially not include any of the changes I suggested.

Author Response

The authors did essentially not include any of the changes I suggested.

Response: We included the changes the reviewer suggested at R1. Information on VLPs in the introduction and results section has been highlighted in yellow on pages 3, 4, and 6.

In the current response, OD values of antibody responses in Figure 5 and Figure 6, which are now Figure 2 and Figure 3 have been replaced by titers as the reviewer commented.   

Reviewer 3 Report

In response to my concerns, the authors have made necessary changes to the ms and most importantly added citations of their missing work. However, the major concern was and still is the novelty of their work. To my understanding, the work has its highlight in the intranasal vaccination which is different from their previous intra-muscular vaccination study. However, in their response, the authors explain that the intranasal single-shot vaccine has already been studied in their previous work [Pharmaceutics]. Therefore, the critique again points back at the novelty of the work to be published as original research rather than a report or review.

The authors have pointed that the study of passive immunization using the VLP immunized mice sera has never been studied in T.gondii. This begs the question then why is it rather not the highlight of the study. This would require a major rewriting of the ms and such a manuscript can be considered for original research. The current emphasis of the ms is not significant enough to warrant an original research publication. Therefore, I propose to authors to either restructure their manuscript to highlight the importance of passive immunity (and include the necessary studies) or submit the current ms as a review article over VLP based vaccines discussing the differences in intra-nasal and intra-muscular, single-dose and booster-dose administration.

Author Response

In response to my concerns, the authors have made necessary changes to the ms and most importantly added citations of their missing work. However, the major concern was and still is the novelty of their work. To my understanding, the work has its highlight in the intranasal vaccination which is different from their previous intra-muscular vaccination study. However, in their response, the authors explain that the intranasal single-shot vaccine has already been studied in their previous work [Pharmaceutics]. Therefore, the critique again points back at the novelty of the work to be published as original research rather than a report or review.

Response: Thank you for the comment. Since passive immunity using the VLP immunized sera in mice has never been studied against T. gondii infection, we restructured the manuscript to highlight the importance of passive immunity as commented. We have newly performed experiments to better support our findings (Figure 2A and Figure 6B).  

The authors have pointed that the study of passive immunization using the VLP immunized mice sera has never been studied in T.gondii. This begs the question then why is it rather not the highlight of the study. This would require a major rewriting of the ms and such a manuscript can be considered for original research. The current emphasis of the ms is not significant enough to warrant an original research publication. Therefore, I propose to authors to either restructure their manuscript to highlight the importance of passive immunity (and include the necessary studies) or submit the current ms as a review article over VLP based vaccines discussing the differences in intra-nasal and intra-muscular, single-dose and booster-dose administration.

Response: Thank you for the comment. We have restructured the manuscript to highlight the importance of passive immunity as commented.       

Round 3

Reviewer 1 Report

  • The idea of restructuring the manuscript to highlight the importance of passive immunity induced by VLP immune sera is interesting. However, it is a little bit strange to start the manuscript with the passive immunization and just later, in section 3.4., explain the VLPs generation and the VLPs immune response generated by VLP immunization. As it is currently restructured it does not flow. It is not necessary to start with passive immunization to highlight the importance of passive immunity induced by VLP immune sera.
  • As suggested by this reviewer authors have included the measurement of IL-4 as a Th2 cytokine. However, authors still consider IL-6 as a Th2 cytokine (in line 304 authors state: Adjuvanted VLPs showed Th2-like cytokines (IL-4, IL-6, IL-10)). Since IL-6 is not a Th2 cytokine, please change this sentence
  • If unimmunized mice were humanely euthanized at 30 dpi it is not correct to make a survival chart (Fig. 6) as at 30 dpi mice are still alive. So please, in line 153, add to the sentence that mice that lost 20% of initial body weight were considered dead and were humanely euthanized as described
  • In Discussion section line 365 authors state: “our study confirmed that passive immunization using anti-T. gondii antibodies can confer protection against T. gondii challenge infection”. This assertion is not correct since mice are not challenged. According to the experimental schedule, mice are first infected and then, 18hs later, passively immunized. Authors should say “our study confirmed that passive immunization using anti-T. gondii antibodies can confer protection against T. gondii infection.”
  • The title in section 3.2 is “Passive immunization with VLPs…..” This is not correct. The passive immunization is not with VLP, it is done with sera from mice immunized with VLP. Please correct this title.

Author Response

Response to Reviewer 1
The idea of restructuring the manuscript to highlight the importance of passive immunity induced by VLP immune sera is interesting. However, it is a little bit strange to start the manuscript with the passive immunization and just later, in section 3.4., explain the VLPs generation and the VLPs immune response generated by VLP immunization. As it is currently restructured it does not flow. It is not necessary to start with passive immunization to highlight the importance of passive immunity induced by VLP immune sera.

  • As suggested by this reviewer authors have included the measurement of IL-4 as a Th2 cytokine. However, authors still consider IL-6 as a Th2 cytokine (in line 304 authors state: Adjuvanted VLPs showed Th2-like cytokines (IL-4, IL-6, IL-10)). Since IL-6 is not a Th2 cytokine, please change this sentence.

Response: The IL-6 has been removed from line 304 and related sections.     

  • If unimmunized mice were humanely euthanized at 30 dpi it is not correct to make a survival chart (Fig. 6) as at 30 dpi mice are still alive. So please, in line 153, add to the sentence that mice that lost 20% of initial body weight were considered deadand were humanely euthanized as described.

   Response: It has been added (line 149).

  In Discussion section line 365 authors state: “our study confirmed that passive immunization using anti-T. gondii antibodies can confer protection against T. gondii challenge infection”. This assertion is not correct since mice are not challenged. According to the experimental schedule, mice are first infected and then, 18hs later, passively immunized. Authors should say “our study confirmed that passive immunization using anti-T. gondii antibodies can confer protection against T. gondii infection.”

Response: It has been corrected as indicated (line 361).

4  The title in section 3.2 is “Passive immunization with VLPs…..” This is not correct. The passive immunization is not with VLP, it is done with sera from mice immunized with VLP. Please correct this title.

Response: It has been corrected ad indicated (line 203).  

Reviewer 2 Report

Thank you for taking into account my considerations.

Author Response

Thank you very much!

Reviewer 3 Report

I thank the authors for taking my concerns seriously and restructuring their work to highlight the importance of passive immunity in T.Gondii infections. I am glad to accept the ms in its current form as the message is clear and I hope the readers will benefit from reading their work.

Author Response

Thank you very much!